# TReSR: A PCR-compatible DNA sequence design method for engineering proteins containing tandem repeats

**James A. Davey** *, **Natalie K. Goto** *

Department of Chemistry and Biomolecular Sciences, University of Ottawa, Ottawa, Ontario, Canada

* jamesa_davey@dfci.harvard.edu (JAD); natalie.goto@uottawa.ca (NKG)

## Abstract

Protein tandem repeats (TRs) are motifs comprised of near-identical contiguous sequence duplications. They are found in approximately 14% of all proteins and are implicated in diverse biological functions facilitating both structured and disordered protein-protein and protein-DNA interactions. These functionalities make protein TR domains an attractive component for the modular design of protein constructs. However, the repetitive nature of DNA sequences encoding TR motifs complicates their synthesis and mutagenesis by traditional molecular biology workflows commonly employed by protein engineers and synthetic biologists. To address this challenge, we developed a computational protocol to significantly reduce the complementarity of DNA sequences encoding TRs called TReSR (for **T**andem **Re**peat DNA **S**equence **R**edesign). The utility of TReSR was demonstrated by constructing a novel constitutive repressor synthesized by duplicating the LacI DNA binding domain into a single-chain TR construct by assembly PCR. Repressor function was evaluated by expression of a fluorescent reporter delivered on a single plasmid encoding a three-component genetic circuit. The successful application of TReSR to construct a novel TR-containing repressor with a DNA sequence that is amenable to PCR-based construction and manipulation will enable the incorporation of a wide range of TR-containing proteins for protein engineering and synthetic biology applications.

## Introduction

The ability to rapidly construct, evaluate, and sequence libraries of protein variants is essential to the workflow employed by protein engineers and synthetic biologists who strive to create proteins and genetic circuits with new and improved properties and functions [1,2]. A useful category of biomacromolecular components can be derived from tandem repeat (TR) amino acid sequence motifs found in approximately 14% of all proteins [3] facilitating an array of both structured and disordered protein-protein and protein-nucleic acid interactions [4,5]. TR sequence motifs encompass a number of modular protein components, including smaller intra-domain motifs forming fibrous structures (e.g., collagen and α-helical coiled-coils) [6,7], intermediate sized motifs (e.g., WD40, leucine-rich, armadillo, ankyrin, Kelch, and HEAT repeat domains) forming elongated solenoid and closed toroid structures [8–13], in addition

**Data Availability Statement:** All relevant data are within the paper and its Supporting information files.

**Funding:** This work was supported by a Natural Sciences and Engineering Research Council

(NSERC, URL: www.nserc-crsng.gc.ca) Discovery Grant (grant number RGPIN-2019-05730) and Discovery Accelerator Supplement (RGPAS-2019-00011) to NKG and an NSERC Postdoctoral Fellowship Award to JAD (funding reference number PDF516965). The funders had no role in study design, data collection and analysis, decision to publish, or preparation of the manuscript.

**Competing interests:** The authors have declared that no competing interests exist.

to larger bead-on-a-string multi-domain motifs [14]. The functional modularity of TR sequence motifs has been exploited in the development of new biotechnologies such as DAR-Pins [15] and TAL effectors [16,17] that allow engineering of specific protein and DNA binding functionality, respectively. Despite their utility and abundance, TR sequence motifs remain underexploited as a class of modular components for the purposes of protein and genetic circuit engineering precisely because they are encoded by repetitive DNA sequences that prohibit the routine application of PCR-based molecular biology techniques [18–21].

Although it is possible to synthesize TR sequences by full-length gene synthesis, the downstream PCR-based manipulations that are routinely employed in protein engineering and synthetic biology workflows will be complicated by the presence of repetitive DNA sequences in these constructs. To circumvent this short-coming, the degenerate nature of DNA codons encoding the canonical amino acids (with the exception of Trp and Met) can be exploited to construct a TR protein encoded by a gene designed to have reduced DNA sequence complementarity, thereby rendering it compatible with downstream PCR-based manipulations. Furthermore, this DNA design strategy should also make it possible to employ the assembly polymerase chain reaction (aPCR) to construct TR-encoding genes more cost-effectively than full-length gene synthesis, since aPCR utilizes oligonucleotide primers as the only template-donating reagent in the reaction [19]. However, the task of redesigning a TR DNA sequence to make it suitable for aPCR is not trivial, as the probability of generating misassembled products increases with the number of primers used in the reaction. Consequently, aPCR approaches in gene synthesis are limited to DNA sequences that have relatively low complementary between non-overlapping segments of primers.

To create a TR-encoding DNA sequence that would be amenable to both aPCR synthesis and PCR-based mutagenesis, we have devised a DNA sequence redesign strategy called TReSR (for **T**andem **Re**peat DNA **Se**quence **R**edesign) to introduce silent mutations that would allow for gene construction by aPCR while preserving the amino acid identity of the translated TR construct (Fig 1). To test this methodology, we designed a novel 178 amino acid residue bead-on-a-string TR protein containing a duplication of the N-terminal DNA binding domain (DBD) of the bacterial repressor LacI [22,23], a construct that has the potential to expand the toolbox of DNA-binding proteins for synthetic biology applications. Application of TReSR allowed for the design of a TR-encoding DNA template having reduced sequence identity (66%) compared to an initial 100% sequence identity between targeted regions of the TRs. This reduction in sequence complementarity enabled synthesis of the full DNA sequence by aPCR and splicing by overlap extension (SOE) [21]. This template was also compatible with PCR-based site-directed mutagenesis, which was used to introduce domain selective triple-mutations designed to specifically bind a variant of the *lac* operator or abolish its DNA-binding activity [24].

The function of the designed repressor was evaluated using a three-component genetic circuit where expression of enhanced green fluorescent protein (eGFP) could be inhibited by expression of our TR repressor construct binding to a unique operator element incorporated in the reporter protein promoter sequence. Measurement of density-normalized culture fluorescence in the absence or presence of an expression-inducing agent for repressor expression demonstrated that only those repressor constructs containing a functional DNA binding sequence in both DBDs could repress expression of eGFP. This genetic circuit was also used to demonstrate that a 19-residue C-terminal truncation of the duplicated DBD, corresponding to the linker helix hinge of LacI (residues 61–89) also served as a functional repressor [25,26]. These results demonstrate the utility of TReSR to manipulate DNA sequence components encoding TRs and create a new DNA binding module that can be used as a repressor in a genetic circuit.

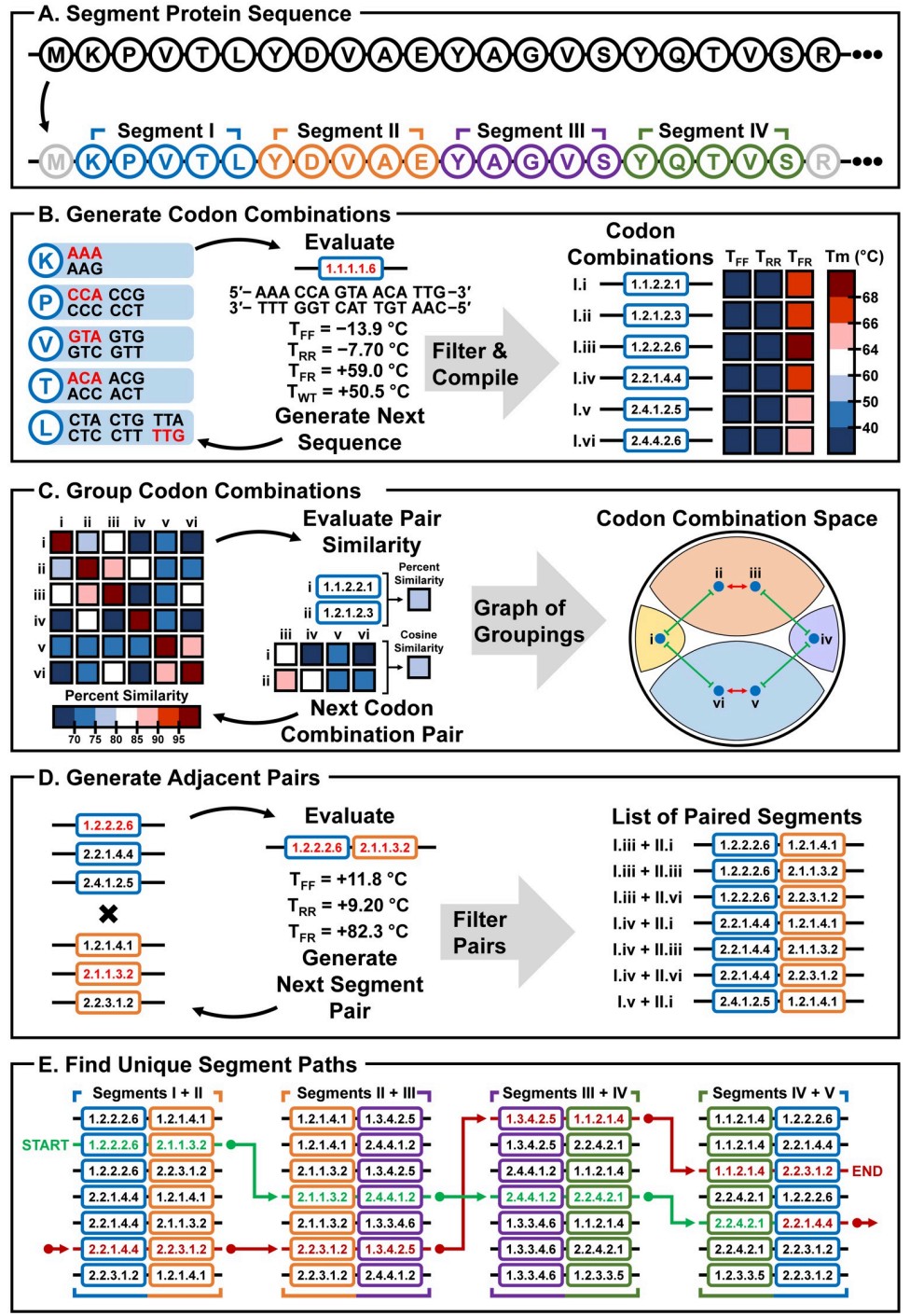

**Fig 1. Overview of the TR DNA sequence redesign strategy implemented in TReSR.** The design strategy presented in this study is schematically outlined for the construction of a TR containing two identical 20 amino acid segments from the N-terminus of the LacI repressor. (A) The TReSR protocol is initiated by dissection of the 20-amino acid target sequence into contiguous 5-residue segments (labelled with upper-case roman numerals) for DNA sequence redesign. (B) This is followed by the generation of a sequence list (with individual sequence entries labelled with lower-case roman numerals) constructed from combinations of synonymous codons that encode the amino acid sequence for each segment. An example codon combination encoding the amino acid sequence for segment I is shown that uses the codons highlighted in red. The label for this codon combination is given by a number for each amino acid that corresponds to the list position for the codon used (e.g., for the sequence shown, the first codon is used for all amino acids except for the last one which used the 6th codon in the list). Melting temperatures (Tm) of all codon

combinations are then calculated using the UNAfold web server24 to provide a measure of homodimerization affinities for the forward ($T_{FF}$) and reverse complement ($T_{RR}$) sequences, along with the Tm of heterodimerization for the forward sequence with its reverse complement ($T_{FR}$) and with the reverse complement of the wild-type sequence ($T_{WT}$). Sequences are then filtered and discarded based on computed hybridization metrics (described in detail in the Materials and Methods), favouring codon combinations that maximize the Tm of heterodimization ($T_{FR}$) while minimizing the Tm of homodimerization (TFF and TRR) and hybridization with the wild-type sequence ($T_{WT}$). (C) The third step of the TReSR protocol assigns codon combinations to groups according to sequence similarity. All pair-wise percent sequence identities are calculated (shown as a heat map for codon combinations (i) to (iv)) and used to identify pairs of codon combinations having high sequence complementarity (e.g., codon combination (i) is similar to (ii) and (iii), and dissimilar to (iv)–(vi)). These are plotted in an interaction graph of codon combination space which is shown for the six codon combinations partitioned into four unique clusters (shaded portions) according to their sequence similarity. (Red arrows indicate codon combinations that share a high degree of percent identity and would therefore be assigned to the same group, while green lines indicate codon combinations that are distinct, and consequently assigned to different groups.) (D) After group assignment, the fourth step involves the assembly of sequences from two adjacent codon combinations from different groups (shown for codon combinations from orange, purple and blue groups from interaction graph). Hybridization metrics are calculated for the joined adjacent segments and then the list of paired codon combinations is filtered (as was done in the second step, B) to eliminate paired segments which are predicted to have problematic homodimerization behaviours (i.e., high $T_{FF}$ and $T_{RR}$). (E) The TReSR algorithm is concluded following a depth-first-search of remaining adjacent codon combinations to identify sequence paths joining contiguous segments. An example TR sequence path is shown with adjacent segment codon combination pairs connected by green arrows for the first domain and continued with red arrows for the path encoding the second domain. A randomly selected sequence resulting from an assembled path is then evaluated as described in the Materials and Methods to confirm that the DNA sequence would be suitable for aPCR construction of the target gene.

## Materials and methods

### Calculation and design of the tandem repeat DNA sequences

The target DNA sequence for the development of our TreSR protocol was a new scDBD containing a TR of two consecutive LacI DBDs (residues 1 through 89). The goal of the DNA sequence redesign procedure was to introduce silent mutations that would allow the duplicated DNA sequence to be constructed via site-selective oligonucleotide assembly by reducing the sequence similarity between the TR-encoding regions while preserving the amino acid identity of the construct. A summary of the TReSR workflow is shown in Fig 1. Segment lengths between 5 and 7 amino acid residues were chosen to reduce the total combinatorial space of silent mutations to be evaluated (Fig 1A). Thermodynamic parameter evaluation was conducted using the DINAMelt two-state melting hybridization application made available through the UNAfold web server [27] to predict melting temperatures (Tm) for homodimeric pairs of forward ($T_{FF}$) and reverse complement ($T_{RR}$) sequences, as well as heterodimerization between the forward and reverse complement ($T_{FR}$) sequences (Fig 1B). Codon segment combinations were then filtered, rejecting segments that form undesired stable homodimers ($T_{FF}$ and $T_{RR}$) or which hybridize with the wild-type LacI sequence ($T_{WT}$), while preferentially selecting for codon combinations that have strong heterodimerization ($T_{FR}$) potentials using a percentile-based threshold calculated for each segment. Specifically, a more stringent 50th percentile was used to set parameter thresholds which minimized potential off-target segment assemblies ($T_{FF}$, $T_{RR}$, and $T_{WT}$) while a less stringent 10th percentile was employed to establish parameter thresholds favouring hybridization with the target segment ($T_{FR}$). The values for these percentile-based thermodynamic parameter thresholds are reported in S1 Table in S1 File. A comparison of similarity between codon combinations encoding the same protein segment was performed by computing pair-wise percent sequence identities (Fig 1C). Codon combinations were grouped according to whether they shared high sequence complementarity (percent sequence identity $\geq$ 80.0%), and whether the codon pair shared similar profiles for percent sequence identity values with respect to the other codon combinations for the segment, as evaluated by a cosine similarity comparison (cos $\geq$ 0.9975). Due to the large number of remaining

codon combinations, the proceeding steps in the TReSR protocol were limited to codon combinations from four randomly selected groups for each segment, excluding all other codon combinations from further consideration (S1 Table in S1 File). All combinations of adjacent pairs of codon combinations were joined, and $T_{FF}$, $T_{RR}$, and $T_{FR}$ values evaluated (Fig 1D) and filtered (S2 Table in S1 File) to discard segment pairs predicted to have high $T_{FF}$, $T_{RR}$ and low $T_{FR}$ values which would prove potentially problematic during aPCR. Again, a percentile-based threshold was employed to discard adjacent codon combination pairs with high homodimerization affinities (20$^{th}$ percentile) while a fixed value for heterodimerization ($T_{FR} = 80.0°C$) was employed to select for an appropriate set of adjacent codon combination pairs to carry forward in the protocol. The TReSR protocol was concluded using a depth-first-search to design the single-chain construct template (Fig 1E), selecting 100 paths which visit distinct codon combinations from different groupings thereby ensuring that the duplicated DNA sequences would be dissimilar and thus amenable to aPCR synthesis. A single path of segments was selected to serve as the TR DNA sequence template reported in S3 Table in S1 File. This TR DNA sequence template was then partitioned into oligonucleotide primers for aPCR synthesis [28]. Refer to the TReSR Computer Code and Documentation section in the Supplementary Information for the Python3.8 computer code and documentation for the TReSR protocol program.

## Construction of the genetic circuit

The pET-11a plasmid (Novagen) was used as the genetic vector to host all three components constituting our genetic circuit. These three components include Cloning Site I whose genetic insert is expressed by the pDBD promoter regulated by the LacI mutant W220F (LacI$_{W220F}$), Cloning Site II serving as the reporter protein expression cassette under the control of the pGFP promoter, and Cloning Site III providing LacI$_{W220F}$ under the control of its native promoter pLacI. This plasmid was transformed and propagated in electrocompetent *Escherichia coli* DH10B [29] via the ColE1 origin with a copy number estimated at 25 to 30 plasmids per cell [30] and AmpR selection marker conferring ampicillin resistance with working concentration of 100 µg·mL$^{-1}$ [31]. Combinations of the pLacI and pLacI$^{Q}$ promoters [32] were paired with the LacI repressor and its variant W220F [33], constructed by successive quick-change PCR reactions. Cloning Sites I and II were incorporated into the pET-11a vector via circular polymerase extension cloning (CPEC) [34] of linear insert cassettes synthesized by aPCR [19] of the pDBD and pGFP promoters and their fusion to the eGFP gene [35] by SOE PCR [21]. Unique restriction enzyme sequences for BamHI and NdeI were introduced to the flanking regions of Cloning Site I, and XhoI and NheI flanking Cloning Site II to enable insertion of gene cassettes at these sites, and both Cloning Sites I and II are flanked by an identical T7 terminator sequence [36]. Insertion of gene cassettes into Cloning Site III were performed using the NdeI restriction sequence belonging to cloning Site I and an EcoRI site 42 BP downstream of the T7 terminator sequence of Cloning Site II.

## Construction of the eGFP and dGFP genes

A copy of *Aequorea victoria* GFP$_{S65T}$ [37] was provided to us by the Chica laboratory, incorporated into the cloning site of the pET-11a. This gene was used as the template to produce eGFP by introducing F64L and H231L mutations by site-directed mutagenesis, yielding eGFP (avGFP$_{F64L/S65T/H231L}$). Notably, our eGFP gene lacks the M1_S2insV insertion mutation corresponding to the NcoI cloning scar found in the originally reported construct [35]. The decoy fluorescent protein (dGFP) used to mimic eGFP expression burden while masking fluorescence output was produced by introducing the R96A mutation [38] into eGFP by quick-change mutagenesis producing avGFP$_{F64L/S65T/R96A/H231L}$.

## Molecular biology reagents and sequencing service

All aPCR, SOE, CPEC, and quick change PCR reactions were performed using Vent DNA polymerase (purchased from New England Biolabs, NEB) and oligonucleotide primers purchased from Eurofins Genomics. Quick change PCR reactions were adapted by replacing the DpnI digestion step with a gel extraction protocol (QIAquick Gel Extraction Kit, Qiagen). Restriction digestion reactions for preparation of vector and insert DNA was processed using BamHI-HF, EcoRI-HF, NdeI, NheI-HF, and XhoI enzymes (NEB). Vector DNA was dephosphorylated using quick cow intestinal phosphatase (QCIP) and ligation reactions conducted using T7 DNA ligase (NEB). Purification of DNA products was done by PCR cleanup (E.Z.N. A Cycle Pure Kit, Omega Bio-Tek) or gel extraction (QIAquick Gel Extraction Kit, Qiagen). Assembled plasmids were transformed into *E. coli* DH10B by electroporation and harvested by miniprep (E.Z.N.A. Plasmid DNA Mini Kit II, Omega Bio-Tek). All culturing was performed in LB Lenox media (BioShop) spiked with 100 μg·mL$^{-1}$ of ampicillin (BioShop). Solid media support was produced by dissolving agar (at a concentration of 15 g·L$^{-1}$, BioShop) in LB (Lenox) liquid media preparation. Induction of Cloning Site I was controlled through isopropyl-β-D-thiogalactopyranoside (IPTG, BioShop) added to achieve a working concentration of 10 mM. Evaluation of *in vivo* reporter protein expression was performed using a SpectraMax M2 plate reader (Molecular Devices) to record optical density ($\lambda_{OD}$ = 600 nm) and fluorescence ($\lambda_{ex}$ = 485 nm, $\lambda_{em}$ = 510 nm, fixed gain = medium, 30 flashes per read) from 100 μL aliquots of cell culture in black-walled and clear-bottomed 96-well format microplates (Greiner). The ColE1 origin and all Cloning Site I, II and LacI genotypes were confirmed by Sanger Sequencing services contracted through Génome Québec (centre d'expertise et de services Génome Québec).

## Construction of the single-chain tandem repeat DNA binding domain repressors

The first single-chain tandem repeat DNA binding domain (scDBD) repressor architecture constructed involved the full-length duplication of the N-terminal LacI DBD sequence (residues 1 through 89), incorporating the triple-mutation DFT (Y17**D**/Q18**F**/R22**T**) producing the non-functional scDBD$_{DFT/DFT}$ construct [24]. This construct was produced in a three-step synthesis where two N-terminal (DBD.N) and two C-terminal (DBD.C) fragments (DBD: residues 1 through 29, and LNK: resides 60 through 89) were first produced by aPCR and gel purified (described in detail in Supplementary Information Section 2). The DBD.N and DBD.C fragment pairs were independently fused to an unchanged intermediate PCR fragment (residues 30 through 59) by SOE PCR and subjected to PCR cleanup, prior to a third SOE reaction producing the full-length construct. This construct was digested (BamHI and NdeI) and gel extracted for insertion into the genetic circuit (prepared by gel extraction of restriction digestion reaction with QCIP, BamHI, and NdeI). The plasmid was harvested by miniprep, followed by confirmation of scDBD$_{DFT/DFT}$ construct identity by sequencing. This template was then subjected to site-directed mutagenesis introducing the functional triple-mutation IAN (D17**I**/F18**A**/T22**N**) at N-terminal and C-terminal duplicated DBDs producing three additional constructs by SOE: scDBD$_{IAN/DFT}$, scDBD$_{DFT/IAN}$, and scDBD$_{IAN/IAN}$. These constructs were similarly inserted into the genetic circuit by digestion, gel purification, and ligation, and harvested by miniprep prior to sequencing. Lastly, C-terminal truncations omitting the duplicated 60 through 89 residue segment of each of the four constructs were produced by PCR: scDBD$_{DFT/DFT/\Delta CT}$, scDBD$_{IAN/DFT/\Delta CT}$, scDBD$_{DFT/IAN/\Delta CT}$, and scDBD$_{IAN/IAN/\Delta CT}$. Likewise, these constructs were inserted into the genetic circuit, harvested by miniprep, and sequenced.

## *In vivo* evaluation of genetic circuit output

The protocol employed to assay and evaluate genetic circuit outputs was adapted from a previous publication assessing the burden imposed upon endogenous expression factors by exogenous genetic circuits [39]. 1 mL LB Lenox pre-cultures were seeded with transformants plated onto a solid LB agar medium, under ampicillin selection. These pre-cultures were grown to stationary phase (37˚C, 300 rpm, 16 hours) and their densities recorded and normalized to 2.6 units ($\lambda_{OD}$ = 600 nm). Density-normalized pre-cultures were passaged into a 2× concentration of LB Lenox (2.6 mL volume) spiked with a 2× concentration of ampicillin (5.2 μL volume). Passaged cultures were distributed in 0.3 mL aliquots into deep 96-well format culture plates across 8 wells (−IPTG: 0.3 mL $H_2O$, +IPTG: 0.3 mL 20 mM IPTG). The resulting culture plate setup allows for twelve separate transformants to be grown in quadruplicate at a 0.6 mL culture volume in the presence and absence of 10 mM IPTG with a starting density of 0.005 units ($\lambda_{OD}$ = 600 nm). Cultures were grown with shaking (37˚C, 300 rpm) and sampled in 100 μL aliquots at the 6, 7, 8, and 9-hour time-points, recording their density and fluorescent output. Linear regression analysis of culture density (Y: $\lambda_{OD}$ = 600 nm) as a function of time (X: hours), fluorescence (Y: $\lambda_{ex}$ = 485 nm, $\lambda_{em}$ = 510 nm, gain = medium, 30 flashes per read) as a function of time (X: hours), and fluorescence (Y: $\lambda_{ex}$ = 485 nm, $\lambda_{em}$ = 510 nm, gain = medium, 30 flashes per read) as a function of optical density (X: $\lambda_{OD}$ = 600 nm) demonstrate that all measurements, regardless of absence or presence of 10 mM IPTG, were linear and recorded at steady-state (S13−.S20 Figs in S1 File). Genetic circuit expression output (*F*) is reported by taking the quotient of fluorescence (*GFP*) and density (*OD*) from each culture measurement (Eq 1).

$$F = \frac{GFP}{OD} \tag{1}$$

All plotted data are reported as the arithmetic average across four measurements, with error bars indicating the standard deviation of the sample. To determine whether changes to genetic circuit outputs are statistically significant, two-tailed homoscedastic *t*-tests were performed with *p*-values reported for populations exhibiting statistically significant difference (*i.e.*, *p*-value $\leq$ 0.001).

## Results

### Overview of DNA sequence redesign protocol

To enable the synthesis of a construct containing TR elements we developed a DNA sequence design protocol called TReSR (for **T**andem **R**epeat **Se**quence **R**edesign) which was implemented as a script run in Python3.8 (S1 File). Using the strategy outlined in Fig 1, TReSR introduces silent mutations into TR sequences to reduce the potential for off-target primer hybridization in PCR reactions, such as those required in aPCR and site directed mutagenesis protocols. The design protocol is conducted in five steps, beginning with the dissection of TR-encoding gene cassette regions into contiguous segments encoding 5 to 7 amino acid residues each (Fig 1A). In the *second* step (Fig 1B), all codon combinations of silent mutations are generated for each segment and melting temperatures (Tm) calculated for the forward ($T_{FF}$) and reverse complement ($T_{RR}$) homodimers, along with that of the forward sequence with its reverse complement ($T_{FR}$) and for the forward sequence with the reverse complement belonging to the wild-type gene ($T_{WT}$). These Tm values are used to exclude sequences prone to homodimerization or hybridization with the wild-type sequence, and include sequences predicted to have strong self-hybridization values. Sequences that do not meet percentile-based

thresholds for these thermodynamic parameters are then discarded before moving to the next step. In the third step (Fig 1C) Tm values are calculated for heterodimers formed between pairs of remaining segment sequences to build an interaction graph and identify a set of compatible sequences for each segment (i.e., segments with unique codon combinations having minimal heterodimerization Tm's). In the fourth step (Fig 1D), unique sequences are paired with adjacent segment sequences and again filtered based on calculated Tm values following the same protocols performed in the second step. The fifth and final step (Fig 1E) involves a depth-first-search joining randomly selected paths from contiguous segment pairs. One assembled path is then chosen at random and primer hybridization parameters evaluated to ensure that the chosen sequence does not have significant off-target hybridization propensity that would complicate its construction by aPCR.

To test the ability of TReSR to design an aPCR-compatible DNA sequence for a designed TR-protein, we chose to construct a single-chain (sc) repressor containing two identical copies of the DBD of the lactose repressor (LacI), called scDBD. Native LacI interacts with DNA as a dimer with each subunit donating an N-terminal DBD that binds one half of the nearly symmetrical *lacO* operator sequence, followed by a linker region and lactose-binding regulatory domain that inhibits DNA binding when bound to 1,6-allolactose or its analogue isopropyl β-D-1-thiogalactopyranoside (IPTG) [23]. Our experimental construct contains two copies of LacI amino acid residues 1–89 organized as a bead-on-a-string tandem repeat. This region of LacI was selected because previously published studies have demonstrated that the duplication of helix-turn-helix domains was sufficient to confer DNA binding capabilities to single-chain repressors [40,41]. Thus, the scDBD repressor construct is designed to bind the operon constitutively to block transcription of the downstream gene. Given the well-characterized suite of DBD-operator sequence combinations that have been identified for this family, this scDBD repressor has the potential to expand the range of transcriptional regulators that can be used in synthetic biology applications.

## Implementation of the TReSR protocol

To make the problem of DNA sequence redesign more tractable, it was necessary to first divide the targeted sequence into smaller segments encoding between 5 and 7 amino acid residues. The choice of a maximum of 7 residues per segment reduced the combinatorial sequence space to a manageable size and also made evaluation of thermodynamic parameters more efficient. This choice of segment length was also convenient for design of a sequence that would be compatible with aPCR, since the DNA encoding these segments would be half the length of a typical oligonucleotide primer needed for this method.

For the redesign of the scDBD TR-encoding sequences, LacI DBD residues 1 through 29 was divided into 5 contiguous segments (labeled A through E), and residues 60 through 89, divided into six contiguous segments (F through K). For each segment (S1 Table in S1 File), all possible codon combinations containing silent mutations were generated, yielding between 128 (segment B) and 3,072 (segment E) different DNA sequences per segment. The UNAFold web server was employed to calculate Tm hybridization values ($T_{FF}$, $T_{RR}$, $T_{FR}$, and $T_{WT}$) for all DNA segments, and the list of segments pruned using percentile-based thresholds ($T_{FF}$ = 0.5, $T_{RR}$ = 0.5, $T_{FR}$ = 0.2, and $T_{WT}$ = 0.5). The values for these thresholds, tabulated in S1 Table in S1 File, pruned approximately 70 to 86% of the total DNA codon combinations belonging to each segment (Fig 1B). For each segment, pairwise percent sequence identity values were calculated for the filtered set of codon combinations. These values were used to construct an interaction graph with vertices, representing individual codon combinations, connected by edges, indicting the percent sequence identity between pairs of codon combinations belonging

to the graph (schematically illustrated in Fig 1C). This graph was used to group codon combinations based on their shared sequence identity. A pair of codon combinations were assigned the same group designation if they shared 80% sequence identity and if they had similar percent identity profiles with the remaining codon combinations in the graph, as determined by a cosine similarity comparison (threshold $\geq$ 0.9975). This grouping procedure produced between 6 (segment B) and 129 (segment H) distinct groupings for the set of eleven protein segments (S1 Table in S1 File). To reduce the total number codon combinations carried forward for the remainder of the TReSR protocol, the space of codon combinations was constrained to those belonging to four randomly selected groups from each segment.

To determine which codon combinations originating from adjacent protein segments were compatible for synthesis by PCR, the thermodynamic parameters ($T_{FF}$, $T_{RR}$, and $T_{FR}$) for DNA sequences constructed from pairs of adjacent codon combinations were compiled, specifically: codon combinations for segment A were paired with those for segment B (A+B), as well as B+C, D+E, F+G, G+H, H+I, I+J, and J+K. This list of adjacent codon combinations was then filtered using the thresholds reported in S2 Table in S1 File (20th percentile for $T_{FF}$ and $T_{RR}$ with a fixed value of 80˚C for $T_{FR}$), reducing the number of paired DNA sequences by approximately 13 to 40%. A depth-first-search was then performed to assemble scDBD DNA template sequences from the set of filtered adjacent segments using contiguous segments belonging to distinct groupings. The resulting template was designed to encode LacI residues 1 through 29 (segments A to E) joined with LacI residues 60 through 89 (segments F to K) to make the N-terminal DBD (DBD.N) joined to a template encoding the same DBD sequence at the C-terminus (DBD.C).

We analyzed the first DNA template produced by TReSR (out of 100 templates generated) and found that 42 and 46 unique silent mutations were introduced into the DBD.N and DBD. C templates across a design space of 59 total codons (Fig 2). The resulting DBD.N and DBD.C templates have reduced sequence identity with each another, measured at 65% between the pair, and the wild-type LacI sequence, recorded at 66% and 63%, respectively. Thermodynamic parameters calculated for segment sequences (S3 Table in S1 File) showed hybridization values that are compatible with aPCR synthesis, with reduced homodimerization Tm (˚C) across all segments ($-58.9 \leq T_{FF} \leq 18.7$ and $-43.7 \leq T_{RR} \leq 19.6$ ˚C). All segment hybridization affinities ($62.5 \leq T_{FR} \leq 74.4$ ˚C) were within the range typically required for annealing and extension steps employed during PCR. An additional advantage of the TReSR-generated sequences are the low hybridization Tm values with the wild-type LacI sequence ($-15.4 \leq T_{WT} \leq 43.7$

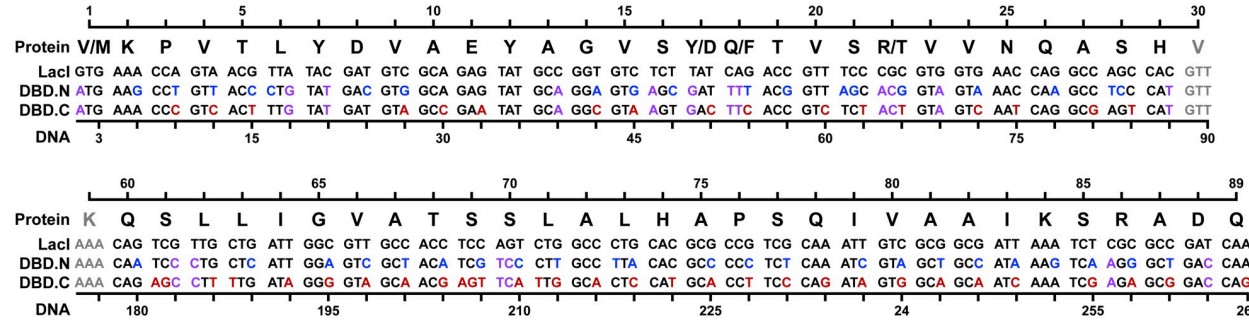

**Fig 2. Single-chain tandem repeat repressor sequence.** A comparison of the redesigned LacI DNA sequences produced by TReSR (residues 1–29 and 60–89) encoding the N- and C-terminal duplicated tandem repeat DBDs (DBD.N and DBD.C respectively), for the single-chain DBD repressor scDBD$_{DFT/DFT}$. Synonymous mutations produced by TreSR are highlighted, with those unique to DBD.N in blue, those unique to DBD.C in red, and those shared by both DBD.N and DBD.C in purple.

˚C) suggesting that downstream PCR manipulation of the scDBD sequence should be possible even with the presence of the LacI gene on the same plasmid.

## Synthesis of the tandem repeat repressor

The template sequence shown in Fig 2 was partitioned into twelve primers for each domain (N.1 – N.12 and C.1 – C.12 for DBD.N and DBD.C, respectively) such that the 3′-termini of all primers were comprised of at least one G/C base-pair and sequence overlap with adjacent primers was designed to ensure efficient assembly of primers (Tm > 64˚C) while limiting length to no more than 44 bases (Table 1). Residues 17, 18, and 22 directing the operator specificity for each DBD are delivered on primers 4 and 5, named according to their triple-mutation identity (DFT: **D**17/**F**18/**T**22 and IAN: **I**17/**A**18/**N**22). According to predictions of thermodynamic parameters shown in Table 1, all primers are expected to adopt linear secondary structures in solution ($\Delta G_F^{72°C}$ and $\Delta G_R^{72°C} > 0.0$ kcal·mol$^{-1}$) preferentially favouring hybridization with their reverse complement sequences ($T_{FR} \geq 75.2$˚C) over formation of undesired homodimers ($T_{FF} \leq 40.7$˚C and $T_{RR} \leq 46.3$˚C). Lastly, a comparison of predicted hybridization affinities between pairs of primers was conducted to ensure successful assembly of the target template DNA sequences for scDBD$_{DFT/DFT}$ and scDBD$_{IAN/IAN}$ (S1 Fig in S1 File). Analysis of predicted hybridization Tm values suggests that all primers will hybridize with sufficient affinity to their adjacent counterparts under reaction conditions employed during PCR ($T_{HYB} \geq 70$˚C) without forming side-products that result from hybridization between pairs of non-adjacent primers.

To create the DNA cassettes encoding scDBD tandem repeat repressor containing either the DFT or IAN set of mutations, PCR reactions were performed to synthesize 4 fragments for each domain, with DBD.N produced by SOE of fragments 1 through 4 and DBD.C produced by SOE of fragments 4 through 7, using the reaction schematic illustrated in S2 Fig in S1 File. Specifically, aPCR was conducted using primers: N.1 through N.6 to produce fragment 2 (encoding LacI residues 1 to 29 for DBD.N), N.7 through N.12 to produce fragment 4 (encoding LacI residues 60 to 89 for DBD.N), C.1 through C.6 to produce fragment 5 (encoding LacI residues 1 to 29 for DBD.C), and DBD.C.7 through DBD.C.12 with T7T.F and T7T.R to produce fragment 7 (encoding LacI residues 60 to 89 for DBD.C). Regions of the scDBD-encoding sequence that were not part of the TR targeted by TReSR were amplified using conventional PCR to make fragments 1, 3 and 6. Successful production of all fragments was supported by agarose gel analysis which all showed a single band at the expected molecular length (S3 Fig in S1 File). SOE was then used to construct the larger fragments encoding the DBD.N domain (composed of fragments 1 to 4) and DBD.C domain (composed of fragments 4 to 7). To construct the full-length tandem repeat constructs scDBD$_{DFT/DFT}$ and scDBD$_{IAN/IAN}$ (sequences provided in S4 Fig in S1 File), fragments encoding DBD.N and DBD.C containing the appropriate triple mutant were joined using a final SOE reaction. Agarose gel electrophoresis demonstrated that all SOE reactions were successful in producing the target fragments (S3 Fig in S1 File).

As intended by the TReSR design protocol, the full-length template containing this TR-encoding sequence could also be used to site-selectively mutate a single DBD without interference from the other DBD in the TR. Moreover, those PCR mutagenesis reactions were performed using DNA templates in a plasmid also carrying the gene for the native LacI repressor. No cross-reactivity with the primers targeting one the TR domains was detected, with only the targeted PCR product being observed by agarose gel electrophoresis (S3 Fig in S1 File) and confirmed by DNA sequencing (S4 Fig in S1 File). This was expected since the TReSR algorithm was designed to exclude sequences that might hybridize with the WT gene (*i.e.*, $T_{WT} \leq$ 43.7˚C for all segment sequences in S3 Table in S1 File). Together, these results demonstrate

**Table 1. Assembly PCR oligonucleotide primers for the TReSR designed tandem repeat repressor.**

| Primer Name | Sequence (5′ → 3′) | Hybridization (°C) | | | Folding (kcal·mol$^{-1}$) | |
|---|---|---|---|---|---|---|
| | | $T_{FF}$ | $T_{FF}$ | $T_{FF}$ | $\Delta G_F^{345K}$ | $\Delta G_F^{345K}$ |
| pDBD.F | CCAGTAGTAGGTTGAGGC | -29.0 | -20.9 | 71.2 | 2.62 | 2.89 |
| pDBD.R | CAGGCTTCATTTTTTTCCTCCTTCTAGTTTAAACAAAATTATTTG | 40.3 | 38.0 | 79.1 | 1.48 | 1.63 |
| N.1 | CTAGAAGGAGGAAAAAAATGAAGCCTGTTACCCTG | -5.0 | -15.2 | 81.2 | 1.68 | 1.48 |
| N.2 | CTCTGCCACGTCATACAGGGTAACAGGCTTCATTTTTTTC | 26.7 | 30.1 | 84.7 | 2.03 | 1.93 |
| N.3 | CCTGTATGACGTGGCAGAGTATGCAGGAGTGAGC | 40.7 | 11.2 | 86.5 | 0.84 | 1.43 |
| DFT.N.4 | CTACCGTGCTAACCGTAAAATCGCTCACTCCTGCATACTCTG | -1.2 | -0.1 | 86.7 | 2.54 | 1.02 |
| IAN.N.4 | CTACATTAGAGACCGTGGCAATGCTCACTCCTGCATACTCTG | 36.6 | 37.3 | 86.7 | 0.94 | 0.93 |
| DFT.N.5 | GATTTTACGGTTAGCACGGTAGTAAACCAAGCCTCCCATG | 28.6 | 36.6 | 85.5 | 1.27 | 1.26 |
| IAN.N.5 | CATTGCCACGGTCTCTAATGTAGTAAACCAAGCCTCCCATG | -20.5 | -3.1 | 86.5 | 1.97 | 1.24 |
| N.6 | CATGGGAGGCTTGGTTTACTAC | -3.1 | -1.6 | 75.2 | 2.29 | 2.61 |
| LacI.N.F | GTAGTAAACCAAGCCTCCCATGTTTCTGCGAAAACGC | 26.2 | 27.6 | 85.7 | 1.82 | 0.22 |
| LacI.N.R | GACTCCAATGAGCAGGGATTGTTTGCCCGCCAGTTG | 25.4 | 29.5 | 88.8 | 1.64 | 1.24 |
| N.7 | CAATCCCTGCTCATTGGAGTCGCTACATCGTC | 10.6 | 14.0 | 84.4 | 1.52 | 1.57 |
| N.8 | GCGTGTAAGGCAAGGGACGATGTAGCGACTCCAATG | 26.3 | 10.6 | 87.8 | 1.57 | 1.72 |
| N.9 | GTCCCTTGCCTTACACGCCCCCTCTCAAATC | -17.0 | -4.3 | 86.7 | 1.72 | 2.26 |
| N.10 | CCTTGACTTTATGGCAGCTACGATTTGAGAGGGGGCGTG | -1.6 | 18.8 | 88.3 | 1.24 | 1.65 |
| N.11 | CGTAGCTGCCATAAAGTCAAGGGCTGACCAAATG | 18.8 | 23.4 | 84.8 | 1.34 | 1.09 |
| N.12 | CATACAAAGTGACGGGTTTCATTTGGTCAGCCCTTGAC | 5.7 | -0.6 | 85.3 | 1.44 | 1.39 |
| C.1 | CAAATGAAACCCGTCACTTTGTATGATGTAG | -4.3 | -2.6 | 77.6 | 2.04 | 1.66 |
| C.2 | GCATATTCGGCTACATCATACAAAGTGACGGGTTTC | -2.6 | -12.4 | 82.7 | 1.66 | 2.04 |
| C.3 | CTTTGTATGATGTAGCCGAATATGCAGGCGTAAG | 29.8 | 32.3 | 81.5 | 1.49 | 1.94 |
| DFT.C.4 | CTACAGTAGAGACGGTGAAGTCACTTACGCCTGCATATTCGG | 33.4 | 35.1 | 86.3 | 1.62 | 0.72 |
| IAN.C.4 | CTACATTGCTCACTGTAGCGATACTTACGCCTGCATATTCGG | 34.4 | 46.3 | 85.7 | 1.38 | 1.10 |
| DFT.C.5 | CTTCACCGTCTCTACTGTAGTCAATCAGGCGAGTCATG | 40.2 | 37.1 | 84.8 | 1.89 | 1.58 |
| IAN.C.5 | CGCTACAGTGAGCAATGTAGTCAATCAGGCGAGTCATG | 47.6 | 39.6 | 85.7 | 1.98 | 1.84 |
| C.6 | CATGACTCGCCTGATTGACTAC | -0.4 | -6.4 | 74.9 | 2.12 | 2.10 |
| LacI.C.F | GTAGTCAATCAGGCGAGTCATGTTTCTGCGAAAACGCG | 30.1 | 31.4 | 86.4 | 1.60 | 0.22 |
| LacI.C.R | GCTCTGTTTGCCCGCCAGTTG | -3.2 | 19.0 | 81.5 | 2.35 | 1.79 |
| C.7 | CTGGCGGGCAAACAGAGCCTTTTGATAGGGGTAGCAACG | 25.1 | 26.8 | 90.0 | 0.93 | 1.57 |
| C.8 | GAACTCGTTGCTACCCCTATCAAAAGG | 8.7 | 1.2 | 78.9 | 1.96 | 1.79 |
| C.9 | GATAGGGGTAGCAACGAGTTCATTGGCACTC | 1.2 | 29.8 | 83.2 | 1.79 | 1.68 |
| C.10 | CTATCTGGGAAGGTGCATGGAGTGCCAATGAACTCGTTG | 30.1 | 22.1 | 87.1 | 1.03 | 0.78 |
| C.11 | CCATGCACCTTCCCAGATAGTGGCAGCAATCAAATCGAG | 15.0 | 17.8 | 87.4 | 1.27 | 1.03 |
| C.12 | CCTATCATTACTGGTCCGCTCTCGATTTGATTGCTGCCAC | 17.8 | 15.0 | 86.7 | 1.39 | 0.41 |
| T7T.F | GAGCGGACCAGTAATGATAGGGATCC | 29.3 | 25.6 | 80.0 | 0.41 | 2.00 |
| T7T.R | GCAGCCGGATCCCTATCATTACTGGTCCG | 34.3 | 30.3 | 85.1 | 1.62 | 0.41 |
| ΔCT.R | GCAGCCGGATCCCTATCATTAACTCGTTGCTACCCCTATCAAAAGG | 34.3 | 30.3 | 88.3 | 2.00 | 1.96 |

that application of the TReSR protocol enabled the design of TR DNA sequence templates suitable for assembly and manipulation by PCR.

## Design of a three-component genetic circuit to evaluate function of scDBD constructs *in vivo*

To evaluate the function of our scDBD constructs, a three-component genetic circuit was designed (Fig 3A) placing expression of the experimental scDBD repressor under inducible

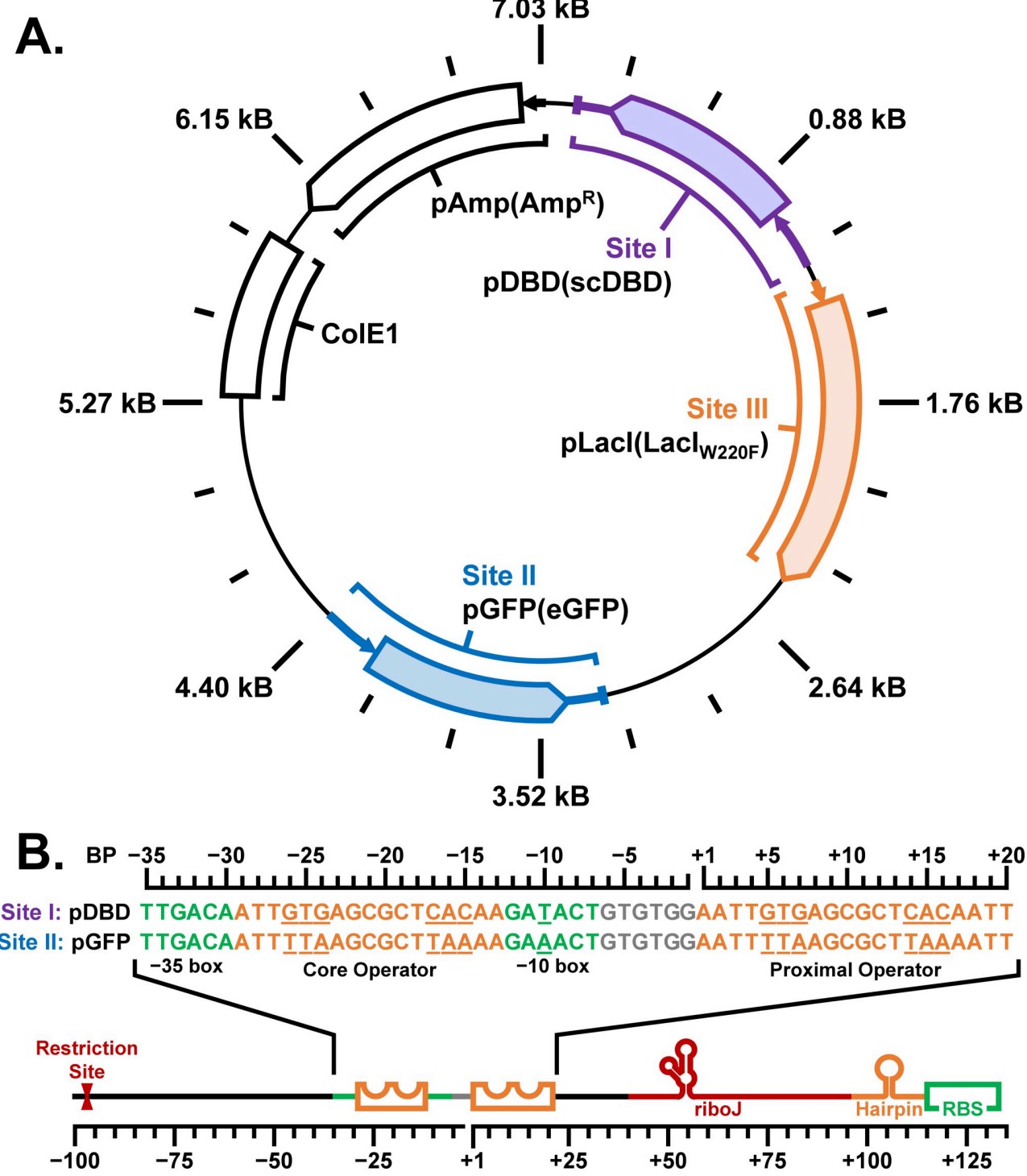

**Fig 3. Architecture of the single-plasmid three-component genetic circuit used to evaluate scDBD repressor constructs.** Plasmid architecture (A) includes the ColE1 origin and ampicillin resistance selection marker (Amp$^R$) in addition to the three genetic circuit components: pDBD(scDBD), pGFP (eGFP), and pLacI(LacIW220F), incorporated at Cloning Sites I, II, and III, respectively. The gene cassettes of Site I (scDBD) and Site II (eGFP) are flanked by identical T7 terminator sequences and near identical pDBD and pGFP promoter sequences, respectively. The promoter regions (B) of Cloning Sites I and II deliver identical riboJ genetic insulator, hairpin, and ribosome binding site (RBS) sequences upstream of the start codon. These promoter cassettes differ by the identity of their −10 box promoter sequence and their operator sequences placed at positions core and proximal along the cassette, with base pair (BP) position indicated relative to the mRNA transcription start site. The operator elements belonging to the pDBD promoter (Site I) incorporate the lacO$^{sym}$ operator sequence responsible for recruiting the LacI$_{W220F}$ repressor expressed by the pLacI promoter from

Cloning Site III. The operator element belonging to the pGFP promoter (Site II) incorporates an operator sequence variant (lacO$^{TTA}$) recognized by the functional scDBD tandem repeat repressor (scDBD$_{IAN/IAN}$: LacI DBD triple mutations Y17$\mathbf{I}$/Q18$\mathbf{A}$/R22$\mathbf{N}$).

control to evaluate its function reported by a cell-based fluorescence assay. We chose to construct our genetic circuit on a single plasmid (sequence provided in S5 Fig in S1 File) since this was expected to reduce its burden on the fitness of its biological hosts by reducing the number of replication origins and selection markers required to propagate and select for the genetic circuit [39]. The genetic circuit contains three components, identified as Cloning Sites I, II, and III, each responsible for delivery of the experimental scDBD repressor, eGFP reporting protein, and LacI repressor protein responsible for regulation of scDBD expression, respectively. Cloning Site III incorporates the gene encoding the W220F variant of the lac repressor (LacI$_{W220F}$) under the control of the constitutive pLacI promoter (pLacI), labelled pLacI (LacI$_{W220F}$). This variant of the LacI repressor was selected after testing a set of plasmids with combinations of repressors (LacI or LacI$_{W220F}$) paired with promoters (pLacI or pLacI$^Q$), confirming the superior ability of pLacI(LacI$_{W220F}$) to repress transcription from the pDBD promoter bearing lacO$^{sym}$ operator sequences in the absence of inducer while simultaneously maximing output expression upon induction (S6–S9 Figs and S4–S8 Tables in S1 File) [33]. The ability of the pLacI(LacI$_{W220F}$) regulatory component to minimize the occurrence of inducer-free (*i.e.*, 'leaky') expression events was required to evaluate scDBD function since the output of the genetic circuit must be reported in the absence and presence of scDBD expression. Details concerning this engineering effort are included in the Supplementary Information section: Engineering and Optimization of the three-Component Genetic Circuit.

Cloning Site I delivers the experimental scDBD repressor constructs that were synthesized by aPCR and SOE (S3 Fig in S1 File) using primers designed by TReSR (Table 1). The scDBD constructs were inserted into Cloning Site I under the control of the promoter pDBD (Fig 3B), outfitted with a pair of lacO$^{sym}$ operator sequences (lacO$^{sym}$: 5′−AATT**GTG**AGCGCT**CAC**AATT −3′), placed at core [39] and proximal [42] positions relative to the RNA polymerase recruitment sequence [43]. Repression of pDBD by LacI$_{W220F}$ is mediated by a specific DBD-operator interacting pair (*i.e.*, the LacI DBD containing the wild-type triple-residue sequence **Y**17/**Q**18/**R**22 is selectively recruited to the lacO$^{sym}$ operator sequence) [24]. This promoter architecture ensures that scDBD expression can be selectively controlled by the addition of IPTG in a dose-dependent manner, minimizing the basal level of expression in the absence of the inducer (S14 Fig in S1 File).

The activity of the genetic circuit is reported using a third component, which delivers the genetically encoded reporter, enhanced green fluorescent protein (eGFP) [35], to Cloning Site II whose promoter, pGFP (Fig 3B), is regulated by the expression of functional scDBD repressor constructs. Specific recruitment of functional scDBD constructs to pGFP is accomplished by employing the DBD triple-mutation Y17**I**/Q18**A**/R22**N** which selectively binds a variant of the symmetric lac-type operator called lacO$^{TTA}$ (sequence 5′−AATT**TTA**AGCGCT**TAA**AATT −3′, with bolded residues indicating site of mutations) [24]. This three-component genetic circuit architecture ensures that repression of pGFP is specifically mediated by functional scDBD repressor without interference from LacI$_{W220F}$ which is incorporated to regulate expression of scDBD constructs. This genetic circuit setup is therefore designed to allow expression of eGFP in the absence of IPTG since expression of scDBD by its promoter (pDBD) is inhibited by LacI$_{W220F}$. Conversely, in the presence of IPTG, LacI$_{W220F}$ dissociates from pDBD, enabling expression of the scDBD repressor candidate which, if functional, would bind to pGFP to repress expression of the reporter protein. Thus, the genetic circuit functions to report on scDBD repressor activity by inverting input induction and output fluorescent signal. To

reduce the potential influence of junction interference on expression levels of eGFP, identical T7 terminator sequences [36] were introduced at the 3′-termini of both Cloning Site I and II coding regions, while upstream promoter elements were outfitted with identical riboJ genetic insulator, hairpin, and ribosome binding site sequences [44,45]. Relative expression levels were measured for pDBD and pGFP promoters in a series of experiments to demarcate the minimum and maximum signal output that can be produced by the genetic circuit in our chosen host expression system, with results described in detail in Supplementary Information section: Expression Controls for the three-Component Genetic Circuit (S10–S13 Figs and S9 Table in S1 File). With this data it was possible to use this single-plasmid genetic circuit to evaluate the function of our designed scDBD repressors in a quantitative manner (S14 Fig in S1 File).

## Evaluation of scDBD repressor function

To evaluate the function of our scDBD repressor, four variants of the genetic circuit were made from a combination of DBDs with the functional IAN (Y17**I**/Q18**A**/R22**N**) and non-functional DFT (Y17**D**/Q18**F**/R22**T**) triple-mutations, incorporated into DBD.N and/or DBD.C domains of our scDBD repressor construct (S15-S22 Figs and S10 Table in S1 File). As native lac repressor binds DNA in a dimeric state, we anticipated that only scDBD repressor constructs incorporating the functional IAN mutation in both DBDs would be able to bind pGFP to repress transcription of the eGFP gene. As shown in Fig 4, a representative sample of the density-normalized fluorescence taken at the 8-hour time point in the presence of 10 mM IPTG resulted in a 5-fold reduction in genetic circuit output signal relative to that obtained for the circuit grown in the absence of IPTG. This repression of eGFP expression by the scDBD repressor was only obtained when both N- and C-terminal DBDs contained the IAN mutation required for recognition of the lacO$^{sym}$ variant operator (lacO$^{TTA}$: G6T/T5/G4A) incorporated in the pGFP promoter. Moreover, the same result was obtained when this combination of scDBDs was truncated (scDBD$_{IAN/IAN/\Delta CT}$) to eliminate the C-terminal copy of the DBD linker region (residues 61 to 89), as only the variant that contained the IAN mutation in both DBDs showed a reduction in fluorescence upon addition of IPTG (3.8 ± 0.2-fold decrease in density normalized fluorescence in the presence of 10 mM IPTG at the 8-hour time point). These results suggest that both scDBD$_{IAN/IAN}$ and its truncated counterpart, scDBD$_{IAN/IAN/\Delta CT}$, act to selectively repress expression from the pGFP promoter without the need for dimerization that is characteristic of the native lac repressor. This demonstration of scDBD repressor function illustrates the ability of TReSR to create new functional proteins containing TR motifs without the need to resort to total gene synthesis.

## Discussion

We chose to demonstrate the utility of TReSR by duplicating a domain-length sequence, in this case the LacI DBD, since this type of TR construct tends to be one of the most difficult to construct by aPCR methods. The repression of eGFP expression via the action of the scDBD$_{IAN/IAN}$ and scDBD$_{IAN/IAN/\Delta CT}$ repressors reported by our genetic circuit demonstrates that even for this challenging system, TReSR was able to create a DNA sequence encoding the TR that allowed its cost-effective assembly by aPCR and SOE. Moreover, the DNA sequence produced by TReSR was also compatible with downstream introduction of mutations by PCR-based site-directed mutagenesis. TReSR design of DNA sequences therefore makes it possible to avoid the well-documented difficulties that are normally associated with manipulating repeating DNA sequences [18]. This presents a significant advantage over other approaches that first independently engineer the function of modular domains and then assemble a final

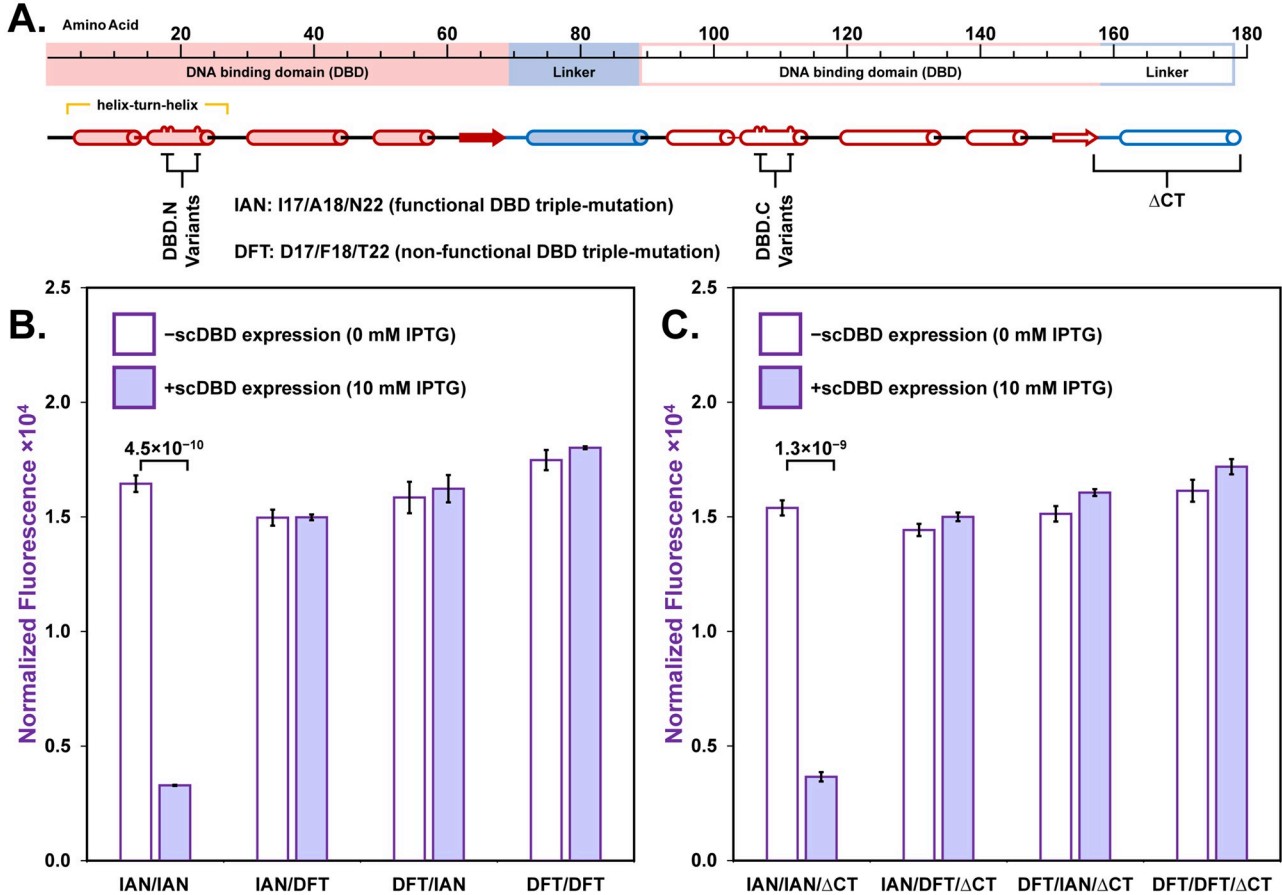

**Fig 4. *In Vivo* evaluation of scDBD repressor function.** Four variants of scDBD repressor protein architecture (A) were inserted into the genetic circuit enabling quantification of repressor function, reported by cell density-normalized fluorescence resulting from expression of eGFP. These four variants are comprised of combinatorial pairs of triple-mutations rendering each duplicated DBD functional (IAN: Y17**I**/Q18**A**/R22**N**) and non-functional (DFT: Y17**D**/Q18**F**/R22**T**). The repressor function of these four variants was evaluated by measuring genetic circuit output when scDBD production is repressed (0 mM IPTG) and when scDBD is expressed (10 mM IPTG) for full-length (B) and C-terminal truncated (C) constructs. Statistically significant differences (*p*-value ≤ 0.001) corresponding to a repression event are indicated (two-tailed homoscedastic *t*-test, n = 4).

TR construct by gene synthesis or DNA ligation. For example, a phage display approach has been employed to identify pairs of zinc finger motifs that could be expressed together as a bead-on-a-string TR-containing protein capable of recognizing DNA sequences in the HIV-1 promoter [46,47]. However, this method cannot be readily applied to TR constructs comprised of modules that do not have function when expressed as individual domains, like the DBDs that were used to engineer the scDBD in this study. As demonstrated using the scDBD triple-mutant variants containing a single functional DBD (*i.e.*, inactivating DFT triple mutations introduced to one of the DBDs), scDBD repressors required two functional DBDs to achieve repression. The application of TReSR to create a new TR-containing protein from the duplication of the LacI DNA binding domain by aPCR demonstrates the simultaneous synthesis and screening of protein constructs incorporating duplicate pairs of modular-TR components. This strategy could be applied to similar protein engineering problems that investigate the scope of modular protein domains hypothesized to be compatible and interchangeable components in TR constructs.

Evaluation of repression activity reported using our genetic circuit demonstrates that the action of this first-generation scDBD TR construct on the pGFP promoter exerts an

approximate 6-fold decrease in expression output relative to the unrepressed pGFP promotor. Comparison of the modest repression activity exhibited by scDBD with results from mutagenesis studies of rheostat positions [48] identified in the linker sequence of LacI [49] suggests suboptimal arrangement and association of the scDBD DNA binding domain pair with the target DNA operator sequence. It is likely that future mutagenesis studies, made possible by the application of our TReSR algorithm, may produce TR constructs with increased repressor activities by optimizing the DNA binding domain pair arrangement to improve scDBD association with its DNA operator.

While our DNA sequence redesign strategy was successfully applied to the generation of a protein containing two domain-length sequences, we anticipate that TReSR should be capable of generating proteins with three repeated domains, since it was possible to perform site-selective PCR-based manipulation of a single DBD domain in a plasmid that contained a total of three DBD sequence copies including the native LacI and the scDBD construct. Moreover, TReSR brought the sequence identity between segments encoding each DBD down to 66%, which can be considered a benchmark for predicting the success of future sequence redesign projects (i.e. aPCR and mutagenesis should be possible if a similar level of sequence identity is obtained from TReSR-designed sequences for other engineered TR proteins). Using this benchmark as a guideline, we anticipate that this will likely be possible for the design of a protein containing four TR domains, since most amino acids are encoded by four degenerate codons. Moreover, for TRs where the repeated sequence is shorter than the domain-length sequences targeted here (e.g. heptad repeat of a leucine zipper), it should be possible to use TReSR to create proteins containing a larger number of TRs.

While the task of computational redesign of DNA sequences to allow PCR-based mutagenesis and manipulation is not unique to this study, this is the first to identify dissimilar DNA sequence fragments prior to assembly of the full-length construct. Previous strategies have been proposed where the targeted sequence is fragmented into oligonucleotides without introducing codon substitutions [28], or where sequence selection is rooted in thermodynamic prediction of oligonucleotide hybridization behaviours [27]. Similar to our strategy, both DNAWorks [50] and Gene2Oligo [51] redesign DNA sequences by computationally evaluating codon substitutions conferring silent mutations which improve the thermodynamic parameters of select oligonucleotides for PCR synthesis. However, neither of these protocols compare DNA sequences of the fragments to reduce the similarity between them, which is critical for the generation of sequences encoding protein TRs. Although our strategy does not include codon usage frequency data when redesigning DNA sequences [52], this parameter could be included in the criteria used to prune codon combination lists (Fig 1B). Alternatively, low frequency codons could be removed from the codon table used in the TReSR calculations, or the effect of low-frequency codons could be mitigated by employing a tRNA-overexpression strategy with cell strains developed for this purpose [53]. This was not required for the scDBDs designed in this study, however, since expression levels of the repressor were sufficient for functional repression.

One of the results arising from our demonstration of TReSR utility is the creation of a new scDBD from the DBD of the lac repressor which was capable of repressing expression from a modified *lacO* promoter. This repressor design is similar to a previously engineered scDBD repressor constructed by duplicating the N-terminal DBD from the bacteriophage 434 cI repressor which recognizes the symmetric 434 operator sequence [40]. DNA sequence recognition of this construct could be predictably altered to produce scDBDs that recognize asymmetric operators to investigate the influence of direct and indirect protein-DNA contacts on repressor-operator binding [54] or to identify cognate and specific protein-DNA interacting pairs [55]. This same strategy for constructing scDBD repressors has also been employed with

the lambda Cro repressor sequence [41]. Our results with the DBD of the lac repressor show that the same strategy can be extended to another well-characterized family of repressors. While the LacI DBD shares a similar helix-turn-helix motif to these bacteriophage repressors, the DNA recognition helix making direct contacts with the operator sequence is oriented in the opposite direction with respect to those bacteriophage repressors [56]. Despite this distinction, our results demonstrate that the same strategy for constructing scDBD architectures from bacteriophage repressors is readily applicable to the LacI DBD and should follow the functional rules defining DBD recognition of operator DNA that had been defined with full-length LacI [24]. In addition, this approach has the potential to be extended to include DBDs belonging to other members of the lac repressor superfamily [22], further increasing the range of promoter sequences that could be recognized.

While TReSR was created for the purpose of redesigning the DNA sequence of a novel TR protein (*i.e.*, the scDBD repressor constructs), this computational strategy also has the potential to be adapted to applications that do not involve TRs. This would involve modification of the TReSR methodology to compare non-identical protein segments, a task that could be facilitated by replacing the percent sequence identity metric (employed in the *third* step of the TReSR protocol, Fig 1C) with calculation of the Tm for hybridization between pairs of DNA sequences. This strategy has the potential to allow the redesign of DNA templates containing problematic regions to make them amenable to PCR-based manipulations by breaking the sequence down into fragments and generating codon combinations with more favorable hybridization parameters. This type of sequence redesign protocol would be particularly useful for mutagenesis of DNA templates in high-throughput procedures (*e.g.*, deep sequencing mutagenesis). The TReSR computational strategy could also be applied to the design and selection of reliable primers for assembly of DNA barcodes used in genotyping large populations of genetic samples. For this application, TReSR could be adapted to compare and select primer combinations that assemble in a defined order to generate unique DNA sequences (*i.e.*, barcodes) appended to amplicons in a single PCR reaction from isolated samples. These samples can then be pooled for next-generation sequencing thus enabling simultaneous sample identification and genotyping, provided that the amplicon length is amenable to the sequencing methodology employed. Similarly, the TReSR protocol could be applied to ribozyme design strategies to design interacting and non-interacting RNA sequences, which would open the door to the engineering of increasingly complex genetic programs and circuitry directing control over gene expression. Overall, the ability to redesign sequences using smaller segments with defined hybridization parameters lies at the core of the TReSR protocol, and offers opportunities for a wide range of potential applications.

## Conclusion

The ability to improve the relative ease of PCR based synthesis and manipulation of TR DNA sequences can alleviate potential experimental complications interfering with the routine utilization of TR sequences for protein engineering applications. To overcome this barrier, we devised and implemented a DNA sequence redesign protocol (TReSR) to construct TR DNA templates that are amenable to assembly and mutagenesis by PCR. TReSR predictions were validated by the construction of a single-chain tandem repeat repressor, created by duplicating the DNA binding domain of LacI. Experimental characterization of repressor construct function using a three-component genetic circuit confirms that this new repressor is functional. The use of TReSR to create TR-containing proteins with DNA sequences that allow aPCR and PCR-based manipulation opens the door to simultaneous screening of both modules and increases the range of TR-containing proteins that can be designed.

## Supporting information

**S1 File. Supplementary Information for TReSR: A PCR-compatible DNA sequence design method for engineering proteins containing tandem repeats.**
(PDF)

**S2 File.**
(ZIP)

## Author Contributions

**Conceptualization:** James A. Davey.

**Data curation:** James A. Davey.

**Formal analysis:** James A. Davey, Natalie K. Goto.

**Funding acquisition:** James A. Davey, Natalie K. Goto.

**Investigation:** James A. Davey.

**Methodology:** James A. Davey.

**Supervision:** Natalie K. Goto.

**Validation:** James A. Davey, Natalie K. Goto.

**Visualization:** James A. Davey.

**Writing – original draft:** James A. Davey, Natalie K. Goto.

**Writing – review & editing:** James A. Davey, Natalie K. Goto.

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
