## [Decision Letter · Decision Letter 0]

27 Feb 2023

PONE-D-23-01509TReSR: A PCR-compatible DNA sequence design method for engineering proteins containing tandem repeatsPLOS ONE

Dear Dr. Natalie,

Thank you for submitting your manuscript to PLOS ONE. After careful consideration, we feel that it has merit but does not fully meet PLOS ONE’s publication criteria as it currently stands. Therefore, we invite you to submit a revised version of the manuscript that addresses the points raised during the review process.

We look forward to receiving your revised manuscript.

Kind regards,

Dharam Singh

Academic Editor

PLOS ONE

Journal Requirements:

"This work was supported by a Natural Sciences and Engineering Research Council (NSERC) Discovery Grant (grant number RGPIN-2019-05730) and Discovery Accelerator Supplement (RGPAS-2019-00011) to NKG and an NSERC Postdoctoral Fellowship Award to JAD (funding reference number PDF516965)."

"This work was supported by a Natural Sciences and Engineering Research Council (NSERC, URL: www.nserc-crsng.gc.ca) Discovery Grant (grant number RGPIN-2019-05730) and Discovery Accelerator Supplement (RGPAS-2019-00011) to NKG and an NSERC Postdoctoral Fellowship Award to JAD (funding reference number PDF516965).

Additional Editor Comments:

Comments as per the reviewers for revision

Reviewers' comments:

Reviewer's Responses to Questions

**Comments to the Author**

1. Is the manuscript technically sound, and do the data support the conclusions?

Reviewer #1: Partly

Reviewer #2: Yes

2. Has the statistical analysis been performed appropriately and rigorously? 

Reviewer #1: Yes

Reviewer #2: Yes

3. Have the authors made all data underlying the findings in their manuscript fully available?

Reviewer #1: Yes

Reviewer #2: Yes

4. Is the manuscript presented in an intelligible fashion and written in standard English?

Reviewer #1: Yes

Reviewer #2: Yes

5. Review Comments to the Author

Reviewer #1: Major points

1. Line 34. The authors state that TR motifs are “unexploited” as a class of molecular components. In fact, there are many types of repeat proteins that have been engineered, including some for commercial purposes (e.g. DARPins and TAL effectors). Maybe the word could be changed to “underexploited,” though even that would be debatable. A similar overstatement is when they say manipulation of TR DNA sequences by PCR is currently “prohibitive” (line 555)

2. The experimental demonstration is fairly limited, just making a single repeat protein of only two units, where many repeat proteins have quite a bit more than two units. Why weren’t known functional repeat proteins like DARPins or TAL effectors engineered to demonstrate the design algorithm works for more than just two-unit repeat proteins? As such, I think the authors can only claim their algorithm works for making two-unit repeat proteins.

3. Changing of codons can result in changes of expression. The authors show that their construct achieves only about 6-fold repression (my estimate based on Fig 4BC). A missing essential control is how much a construct with an exact repeat of the wildtype sequence would repress (note that there are a lot of control experiments in the supplementary, but I had a hard time understanding the authors’ nomenclature and I think all these control experiments were performed with tandem repeats that were not repetitive in the DNA sequence). I would expect it to be much more that 6-fold based on the typical properties of the lac repressor, but I don't know for sure. One reason for poor performance would be poor translation due to the DNA sequence (i.e. non-optimal codons or regions of the gene). Although they say codon usage information can be incorporated into their method, they did not do so in this study (even so, codon optimization for expression is much more complicated than just choosing the often-used codons for that organism). If the authors algorithm produces a tandem duplicate gene that has poor expression, that is a caveat to the authors’ method. But if the authors construct represses about as well as the non-optimized gene, then the authors method would be more useful.

Minor points

4. Line 242. ABS is not the best abbreviation for what is being measured. What is being measured is optical density (OD) not absorbance (i.e. the physical phenomena of why the measurement increases with cell density is light scattering, not absorbance).

5. Variables should be italicized in text (e.g. line 244 the variables in the equation should be italicized and line 247 t-test and p-value, the t and p should be italicized.)

Reviewer #2: The study is well organized and clear in its objectives. Authors have done a commendable work. The tool described here will further help researchers in protein engineering. It will have important contribution in synthetic biology area. The manuscript can be accepted for publications with some minor comments and suggestions.

1. How the size of the proteins will effect on the total protein expression in this case?

2. How much important is sequence length?

3. Have the authors tested repressors other than LacI?

corrections: Please supply high quality images for all figures as they become blurred upon zooming.

6. PLOS authors have the option to publish the peer review history of their article (what does this mean?). If published, this will include your full peer review and any attached files.

Reviewer #1: No

Reviewer #2: **Yes: **VIRENDER KUMAR

---

## [Author Response · Author response to Decision Letter 0]

21 Mar 2023

Response to editor comments was provided in the cover letter, and response to the reviewers was provided in a separate file.

---

## [Decision Letter · Decision Letter 1]

29 Mar 2023

TReSR: A PCR-compatible DNA sequence design method for engineering proteins containing tandem repeats

PONE-D-23-01509R1

Dear Dr. Goto,

We’re pleased to inform you that your manuscript has been judged scientifically suitable for publication and will be formally accepted for publication once it meets all outstanding technical requirements.

Kind regards,

Dharam Singh

Academic Editor

PLOS ONE

Additional Editor Comments (optional):

Reviewers' comments:

Reviewer's Responses to Questions

**Comments to the Author**

1. If the authors have adequately addressed your comments raised in a previous round of review and you feel that this manuscript is now acceptable for publication, you may indicate that here to bypass the “Comments to the Author” section, enter your conflict of interest statement in the “Confidential to Editor” section, and submit your "Accept" recommendation.

Reviewer #1: All comments have been addressed

Reviewer #2: All comments have been addressed

2. Is the manuscript technically sound, and do the data support the conclusions?

Reviewer #1: Yes

Reviewer #2: Yes

3. Has the statistical analysis been performed appropriately and rigorously? 

Reviewer #1: Yes

Reviewer #2: Yes

4. Have the authors made all data underlying the findings in their manuscript fully available?

Reviewer #1: Yes

Reviewer #2: Yes

5. Is the manuscript presented in an intelligible fashion and written in standard English?

Reviewer #1: Yes

Reviewer #2: Yes

6. Review Comments to the Author

Reviewer #1: The authors have addressed my comments.

The authors have addressed my comments.

The authors have addressed my comments.

Reviewer #2: All concerns are addressed. The authors have now answered all the queries and incorporated the suggestions. The manuscript can be accepted.

7. PLOS authors have the option to publish the peer review history of their article (what does this mean?). If published, this will include your full peer review and any attached files.

Reviewer #1: No

Reviewer #2: **Yes: **Virender Kumar

---

## [Editor Report · Acceptance letter]

3 Apr 2023

PONE-D-23-01509R1 

TReSR: A PCR-compatible DNA sequence design method for engineering proteins containing tandem repeats 

Dear Dr. Goto:

I'm pleased to inform you that your manuscript has been deemed suitable for publication in PLOS ONE. Congratulations! Your manuscript is now with our production department. 

Kind regards, 

on behalf of

Dr. Dharam Singh 

Academic Editor

PLOS ONE